# An end-to-end implicit neural representation architecture for medical volume data

**Armin Sheibanifard[1], Hongchuan Yu[1]\*, Zongcai Ruan[2], Jian J. Zhang[1]**

**1** NCCA, Bournemouth University, Poole, United Kingdom, **2** Key Laboratory of Child Development and Learning Science, South-East University, Nanjing, China

\* hyu@bournemouth.ac.uk

**Data Availability Statement:** The data and codes supporting the findings of this study are openly available at GitHub and can be accessed at https://github.com/asheibanifard/EndtoEndCompression.

## Abstract

Medical volume data are rapidly increasing, growing from gigabytes to petabytes, which presents significant challenges in organisation, storage, transmission, manipulation, and rendering. To address the challenges, we propose an end-to-end architecture for data compression, leveraging advanced deep learning technologies. This architecture consists of three key modules: downsampling, implicit neural representation (INR), and super-resolution (SR). We employ a trade-off point method to optimise each module's performance and achieve the best balance between high compression rates and reconstruction quality. Experimental results on multi-parametric MRI data demonstrate that our method achieves a high compression rate of up to 97.5% while maintaining superior reconstruction accuracy, with a Peak Signal-to-Noise Ratio (PSNR) of 40.05 dB and Structural Similarity Index (SSIM) of 0.96. This approach significantly reduces GPU memory requirements and processing time, making it a practical solution for handling large medical datasets.

## 1 Introduction

Medical visualisation commonly involves volumetric medical data such as CT, MRI, PET scans, and confocal spectral microscopy images. This technique is essential in clinical practices across various biomedical disciplines, like radiology, nuclear medicine, surgery planning, and nearly all neuroscience sub-fields. However, the generated volume data often reaches enormous sizes. The generated data often becomes very large, sometimes reaching terabyte-scale. For instance, biological volumetric datasets that capture microscale details of cells or tissues are commonly produced [1–5]. The emerging challenges lie in organising, storing, transmitting, manipulating, and rendering such terabyte-scale volume data.

Recent advances in deep neural networks have led to their rapid application in medical imaging [6–8]. In particular, implicit neural representations have become an approach for compressing volumetric medical images by storing the parameters of trained neural networks instead of explicit voxel data such as SIREN [9]. However, the compression rate is often limited and volumetric data still require considerable memory, especially GPU memory. This results in high memory demands and longer training times for deep learning applications. In addition, there is currently a scarcity of research addressing these specific challenges.

**Funding:** This research was partially supported by the EU Horizon Project-ACMod (No. 101130271). Zongcai Ruan was supported by STI2030-Major Projects of China (2021ZD0204002). The funders had no role in study design, data collection and analysis, decision to publish, or preparation of the manuscript.

**Competing interests:** The authors have declared that no competing interests exist.

To address these challenges, this paper presents an End-to-End architecture that improves compression rates and reduces GPU memory usage, based on our previous work [10]. The proposed architecture consists of three key modules: a downsampling module, an Implicit Neural Representation (INR) module, and a 3D Super-Resolution (SR) module (e.g., [11]). The downsampling module reduces data size, enabling the INR module to represent the volume using a compact deep neural network. The SR module then reconstructs the original high-resolution volume from the INR module output. This architecture reduces memory needs and allows for more efficient neural network training. The main challenge lies in achieving a high compression rate and minimal reconstruction loss. To address this, we propose a trade-off point method that optimises the configuration of each module to achieve peak performance. This approach can be generalised to a wide range of deep network designs. Our key contributions include:

- We propose an End-to-End architecture with three computational modules, designed to optimise volumetric data compression by achieving a high compression rate while maintaining superior reconstruction quality and minimising GPU memory consumption.

- We introduce a trade-off point method to determine the optimal configuration for the proposed End-to-End architecture, balancing key performance metrics such as compression rate and reconstruction quality.

The rest of the paper is structured as follows. Section 2 briefly reviews related work. Section 3 presents the proposed architecture and the trade-off point method. Section 4 presents experimental results and analysis. Finally, Section 5 concludes our work.

## 2 Background and relevant literature

In our previous work [10], we developed an architecture that leveraged existing pre-trained deep networks to decrease the volume data size. The basic idea is to transform volume data into an implicit neural network representation, such as SIREN [9], to compress the data while maintaining reconstruction accuracy. However, pre-trained deep networks often struggle to generalise well, especially with medical volume data. Many pre-trained Super-Resolution deep networks require fine-tuning for different medical datasets. A "one-size-fits-all" approach does not work, since each dataset has its own characteristics. The existing deep networks do not generalise well to diverse volume data. Therefore, this paper aims to train an end-to-end deep network, rather than simply piecing together multiple pre-trained networks.

### 2.1 Implicit neural representation

Representing 3D geometry for rendering and reconstruction involves trade-offs across fidelity, efficiency, and compression capabilities. The DeepSDF model [12] uses a continuous Signed Distance Function (SDF) to represent shapes. Another approach [13] employs an encoder-decoder neural architecture for lossless compression. However, this method has a high inference time due to explicit optimisation requirements.

MedZip [14] proposes a lossless compression technique employing Long Short-Term Memory (LSTM) for volumetric MRI and CT. NeRF [15] presents a notable method for synthesising new views of a volumetric scene through implicit neural representation as a continuous function. However, it is outperformed by SIRENs [9] due to its time consumption. [16] presents a 3D representation technique to reduce memory usage by predicting an occupancy function for a continuous volume. COIN [17] applies a multi-layer perceptron (MLP) to implicit neural network compression by encoding geometric inputs. However, it

demonstrates inferior performance compared to state-of-the-art compression methods. INR-GAN [18] applies a GAN model to multi-scale Implicit Neural Representations (INRs) but struggles with artefacts when dealing with high-frequency features. NeRP [19] introduces a novel approach to generate a computational image from sampled sensor data. However, dealing with sparsely sampled images encounters additional hurdles due to limited data points. Unlike previous deep learning methods for image reconstruction, NeRP leverages both the internal structure of an image prior and the physics governing sparsely sampled measurements to represent the entire subject.

## 2.2 Super-resolution techniques

Numerous techniques leveraging convolutional neural networks (CNNs) have demonstrated exceptional performance in image super-resolution (SR). The pioneering work of SRCNN [20] introduced CNNs to SR by learning a non-linear mapping from low-resolution to high-resolution images with only three convolution layers. CNN-based methods illustrated their impressive performance in SR. Still, they became impractical when taking into account constraints on time and memory resources [21–30]. SRNO [11] designed for continuous super-resolution tasks. It treats each image as a function and learns a mapping between finite-dimensional function spaces, enabling it to train and generalise across various discretisation levels. Experiments demonstrate that SRNO surpasses other arbitrary-scale super-resolution methods in terms of both performance and computational time, particularly excelling in capturing global image structures, which is important in medical imaging.

Table 1 highlights the gaps between the proposed method and four state-of-the-art models —SIREN [9], MedZip [14], NeRF [15], and COIN [17]—across several key metrics: high compression rate, low GPU memory consumption, high reconstruction quality (PSNR > 40), good visual similarity (SSIM > 0.9), scalability to large datasets, fast training time, adaptability to medical imaging, and handling high-frequency features. The proposed method addresses several limitations of existing models, particularly in achieving high compression rates and excellent reconstruction quality, while maintaining efficiency in GPU memory usage and adaptability to medical imaging tasks.

## 3 Methodology

In this section, we first present the end-to-end architecture and then introduce the trade-off point approach to evaluate the proposed architecture in terms of compression efficiency and reconstruction accuracy.

**Table 1. Identifying gaps in state-of-the-art models compared to the proposed method.**

| Feature/Metric | Proposed Method | SIREN [9] | MedZip [14] | NeRF [15] | COIN [17] |
|---|---|---|---|---|---|
| High Compression Rate | ✓ | ✓ | | | ✓ |
| Low GPU Memory Consumption | ✓ | ✓ | | | |
| High Reconstruction Quality (PSNR > 40) | ✓ | | ✓ | ✓ | |
| Good Visual Similarity (SSIM > 0.9) | ✓ | ✓ | | ✓ | |
| Scalable to Large Datasets | ✓ | | | ✓ | |
| Fast Training Time | ✓ | ✓ | | | |
| Adaptability to Medical Imaging | ✓ | | ✓ | | |
| Handles High-Frequency Features Well | ✓ | ✓ | | ✓ | ✓ |

### 3.1 Proposed end-to-end architecture

Our end-to-end architecture, shown in Fig 1, is composed of three core modules: Downsampling, Implicit Neural Representation (INR), and Super-Resolution (SR). The Downsampling module does not require training. We need to train the INR and SR modules in an end-to-end way. We employ a $L_1$ loss function to evaluate reconstruction quality here. In the following sections, we will explain each module individually.

**3.1.1 3D downsampling module.** Given a high-resolution volume of *x*, this module aims to acquire its low-resolution counterpart y. The relationship between x and y can be modelled as follows,

$$y = \mathbb{F}_{LR}^{-1}\mathbb{D}\mathbb{F}_{HR}x + n \tag{1}$$

where, $\mathbb{F}_{HR}$ is the FFT operator for the high-resolution regime, $\mathbb{F}_{LR}^{-1}$ is the inverse FFT operator for the low-resolution regime, $\mathbb{D}$ is the low-pass operator on the frequency domain, and *n* is the noise. Fourier Transform technique is widely employed in medical imaging [31]. We hope to point out that the operator $\mathbb{D}$ in the frequency domain is both controllable and easy to implement. In our case, it effectively generates low-resolution volumes at downsampling scales of $\times\frac{1}{2}$, $\times\frac{1}{4}$, and $\times\frac{1}{8}$. Additionally, it can be noted that this module does not need training.

**3.1.2 3D implicit neural representation (INR).** The INR module harnesses the capabilities of implicit neural networks to efficiently encode volumetric data. Specifically, using INR for low-resolution volumes helps prevent memory overflow. Unlike conventional explicit representations, INRs depict the volume as a continuous function that maps spatial coordinates to voxel intensity values. This enables a concise representation that can be readily adjusted to different levels of detail. Drawing inspiration from recent breakthroughs in implicit neural representations, we employed a multi-layer perceptron (MLP) architecture with periodic activation functions (i.e., SIREN [9]) to effectively capture the intricate structures within the volumetric data.

**3.1.3 3D super resolution (SR) module.** The SR module employs the super-resolution model, SRNO [11]. SRNO model utilises deep learning to learn intricate transformations from low-resolution to high-resolution data. Beyond enhancing resolution, SRNO models frequently possess intrinsic denoising abilities, resulting in cleaner and clearer images. Compared to other super-resolution techniques, SRNO models can produce images with fewer artefacts, such as ringing and blurring [11]. Moreover, the number of channels in the attention structure can significantly influence the SRNO model's performance. Thus, we regard it as a hyperparameter of the SRNO models and evaluate the SRNO by it.

### 3.2 Trade-off point approach

To achieve an overall optimal performance for our proposed end-to-end architecture, we propose a metric system to measure overall performance and further determine the optimal setting for each module accordingly. This design method is called the Trade-off Point Method. Our metric system includes four measurements: Peak Signal-to-Noise Ratio (PSNR), Structural Similarity Index (SSIM), Bitrate, and Compression Rate (CR) as below. PSNR provides a measure of pixel-level accuracy by calculating the ratio of signal power to noise power, yet it often does not correspond to human visual perception. In contrast, SSIM assesses perceptual quality by comparing luminance, contrast, and structure, but may overlook precise pixel-wise errors. Recognising the limitations of using PSNR or SSIM alone for performance measurement, we combine both metrics to evaluate image quality thoroughly.

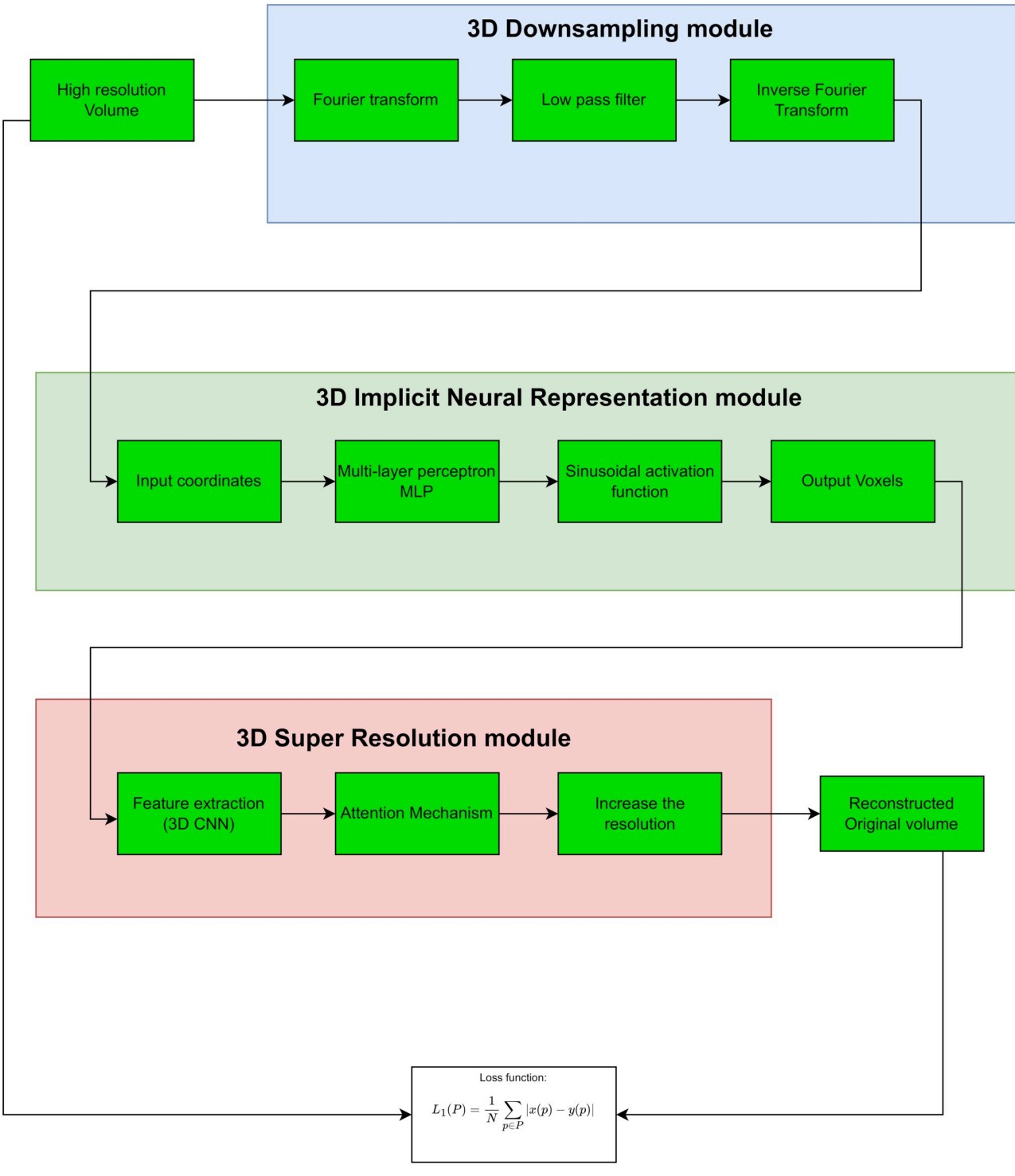

**Fig 1. Workflow of the proposed end-to-end architecture, including downsampling, implicit neural representation (INR), and super-resolution (SR) modules.**

### 3.2.1 Metric definition.

- **Peak Signal to Noise Ratio(PSNR)** is a metric used to measure the quality of a reconstructed or compressed signal compared to the original signal. It is expressed in decibels (dB) and is calculated using the following formula:

$$PSNR = 10 \cdot \log_{10}\left(\frac{MAX^2}{MSE}\right) \tag{2}$$

where: MAX is the maximum possible pixel value of the image (e.g., 255 for an 8-bit image), and MSE is the Mean-Squared Error between the original and reconstructed images.
A high PSNR value indicates a high-quality reconstruction, as it signifies that the reconstructed signal is closer to the original signal in terms of fidelity.

- **Structural Similarity Index Measurement(SSIM):** The Structural Similarity Index Measurement(SSIM) is a metric to assess the similarity between a reference image (original) and a distorted or processed image. SSIM quantifies similarity by considering three key components: luminance, contrast, and structure. SSIM is defined as,

$$SSIM(x, y) = \frac{(2\mu_x\mu_y + C_1)(2\sigma_{xy} + C_2)}{(\mu_x^2 + \mu_y^2 + C_1)(\sigma_x^2 + \sigma_y^2 + C_2)} \tag{3}$$

where: $\mu_x$ and $\mu_y$ are the means of the original and distorted images, respectively, $\sigma_x^2$ and $\sigma_y^2$ are the variances of the original and distorted images, respectively, $\sigma_{xy}$ is the covariance of the original and distorted images, $C_1$ and $C_2$ are small constants added for numerical stability. The SSIM value ranges from -1 to 1, with 1 indicating perfect similarity. High SSIM values indicate high similarity between the images, while low values suggest more significant differences or distortions.

- **Bitrate:** Bitrate is a metric used in digital imaging to quantify the amount of data assigned to each pixel in a raster image. Bpp indicates the level of detail or precision in representing colour or intensity information for each pixel. High Bpp values typically result in high image quality but large file size, while low Bpp values lead to low quality but small files. It is computed as,

$$Bitrate = \frac{Total\ bits}{Total\ pixels} \tag{4}$$

In greyscale images, each pixel is represented by a single channel (e.g., luminance). Bpp is degraded as,

$$Bitrate = \frac{Bit\ depth}{1} \tag{5}$$

When compression techniques are applied, the Bitrate measures the density of the pixel value of the image to assess the trade-off between image quality and file size. High Bitrate values generally result in high-quality but large image files, while low Bitrate values lead to more aggressive compression and small files but with potential quality loss.

- **Downsampling Scale (DS):** Let $D_x$, $D_y$, and $D_z$ be the original dimensions of the 3D image stacks in a $(x, y, z)$ coordinate system, respectively; and the new dimensions be $(d_x, d_y, d_z)$

after downsampling. The DS $(s_x, s_y, s_z)$ is defined as,

$$d_x = \frac{D_x}{s_x}, d_y = \frac{D_y}{s_y}, d_z = \frac{D_z}{s_z}$$

We may simply set $(s_x, s_y, s_z)$ identically.

- **Number of the neurons in SIREN (SN):** With SIREN's layer count set at 3, each layer contains an identical number of neurons. We adjust the neuron count per layer from 30 to 230, using this to represent SIREN's size.

- **Number of Channels (NC):** We incorporate the 3D version of SRNO into the SR module. The cornerstone of a super-resolution network lies in its feature extractor. Existing super-resolution models possess their own topologies for their feature extractors. The number of Channels indicates the feature extractor's size, thereby reflecting the complexity of the super-resolution network. This complexity is particularly influenced by the downsampling scale within our proposed architecture, leading to a significant increase in channel numbers due to the abundance of volume data. To minimise the size of the SR module in our proposed architecture, we initially assess the performance of the SR module with different sizes of attention mechanisms and fully connected layer submodules, after which we fix the topologies and sizes of these two submodules. However, the channel number of the feature extractor remains adaptable to accommodate varying reconstruction accuracy requirements.

- **Compression Rate (CR):** The CR refers to the ratio of the compressed data's size over the uncompressed data's size. A high compression rate indicates an efficient compression process, as it signifies a remarkable reduction in data size. It is defined as,

$$CR = \left( 1 - \frac{\text{Size of the network}}{\text{Size of Uncompressed Data}} \right) \times 100\% \tag{6}$$

In this paper, we define the size of a deep network by its weight count and the size of a volume by its voxel number.

**3.2.2 Trade-off settings.** To find the trade-off settings for the individual modules, we first apply the metrics of PSNR, SSIM, and CR defined in the above section separately to a specific volume of data concerning three dimensions: DS, NC, and SN. The different combinations of DS, NC, and SN result in different measurements, which are stored in a 3D array, as shown in Fig 2. We need to balance the performance of (PSNR, SSIM, and CR) associated with the combination of three dimensions (DS, NC, SN) to determine the trade-off point for our end-to-end architecture. This may be described as,

$$\begin{cases} \min_{(x,y,z)\in 3DA} \left( \frac{1}{\text{PSNR}} + 1 - |\text{SSIM}| + (1 - \text{CR}) \right) \\ \\ \text{subject to} \begin{cases} c_1: & x - \text{DS}_{\text{max}} = 0 \\ \\ c_2: & y - \text{NC}_{\text{min}} = 0 \\ \\ c_3: & z - \text{SN}_{\text{min}} = 0 \end{cases} \end{cases} \tag{7}$$

where, 3DA denotes the 3D array with 3 dimensions, DS, NC, SN, and $DS_{max}$ denotes the given maximum value for DS, and others have a similar definition. Applying the Augmented

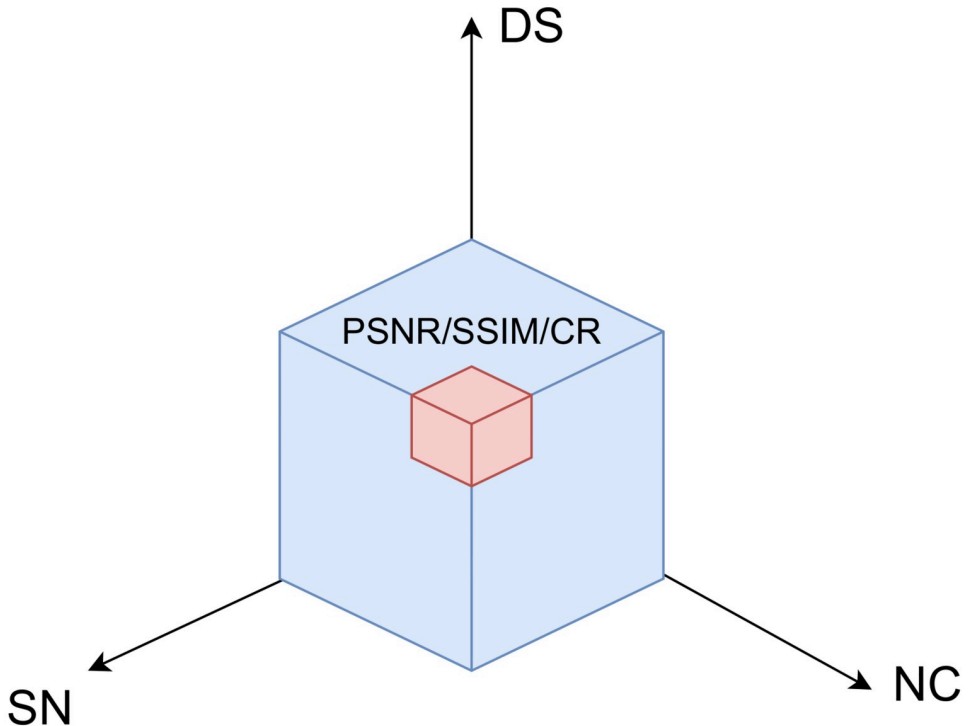

**Fig 2. Illustration of the data structure in the context of the metrics, PSNR, SSIM and CR, according to the DS, NC and SN dimensions.**

Lagrangian method here yields,

$$\text{TradeOff} = \operatorname*{argmin}_{(x,y,z)\in 3DA}\left(\frac{1}{\text{PSNR}} + 1 - |\text{SSIM}| + (1 - \text{CR}) - \sum_{i=1}^{3}\alpha_i c_i + \frac{1}{2}\beta\sum_{i=1}^{3}c_i^2\right) \qquad (8)$$

where $\alpha$ are Lagrange factors and $\beta$ is the penalty parameter. The resulting (x,y,z) is called the trade-off point. To visualise it, we compute the marginal distributions concerning three dimensions separately on 3DA as below,

$$\begin{cases} PSNR(x \sim 3DA(DS)) = \sum_{(y,z)\sim 3DA(NC,SN)} PSNR(x,y,z) \\ SSIM(x \sim 3DA(DS)) = \sum_{(y,z)\sim 3DA(NC,SN)} SSIM(x,y,z) \\ CR(x \sim 3DA(DS)) = \sum_{(y,z)\sim 3DA(NC,SN)} CR(x,y,z) \end{cases} \qquad (9)$$

There are a total of three sets of marginal distributions. Each set illustrates the PSNR bounds, SSIM bounds, and CR bounds concerning the scale at each dimension specified by the trade-off point, one after another. Theoretical equivalence is expected among these three sets of PSNR, SSIM and CR bounds at the trade-off point. The trade-off point indicates the tolerance of the proposed architecture in three dimensions at an expected PSNR, SSIM and CR bounds level. The area delimited by the trade-off point intuitively and quantitatively illustrates the proposed architecture's performance.

## 4 Materials and experimental results

Our experiments can be categorised into two parts. The first part aims to justify the selection of each module in our proposed end-to-end architecture. The second part involves applying the trade-off point method to determine an optimal architecture that balances various considerations.

### 4.1 Data and implementation setup

The dataset comprises 750 multi-parametric magnetic resonance images (mp-MRI) collected from patients diagnosed with either glioblastoma or lower-grade glioma [32]. We select T2 Fluid-Attenuated Inversion Recovery (FLAIR) 3D scan from a random patient with the size of 155 x 240 x 240. The implementation of our architecture starts with a high-resolution 3D volumetric input, such as a medical scan, denoted as $x$. Initially, the input volume undergoes normalisation, scaling the voxel values to a range between 0 and 1. To streamline computations, the volume is segmented into smaller patches, each measuring $64 \times 64 \times 64$. Patches with 70% or more non-zero voxels containing more information are classified as High-Resolution (HR) patches. From these, one HR patch is selected as the high-resolution input for further processing.

Once the data are prepared, the 3D Downsampling module applies a Fourier Transform to convert the high-resolution volume from the spatial domain to the frequency domain. A low-pass filter is then used to eliminate high-frequency components, thereby reducing resolution. This removal process is crucial in medical imaging, as it decreases the data size while preserving essential information, ultimately easing the model processing load. The Inverse Fourier Transform reverts the data to the spatial domain, yielding a low-resolution version of the original volume.

Next, the downsampled volume is processed through the 3D Implicit Neural Representation (INR) module. Here, a Multi-Layer Perceptron (MLP) utilising Sinusoidal Activation Functions (SIREN) maps input coordinates to output voxel intensities, enabling the neural network to represent complex structures as continuous functions. These functions are then converted into voxel intensities.

Following this, the 3D Super-Resolution (SR) module employs a 3D Convolutional Neural Network (CNN) for feature extraction, incorporating an Attention Mechanism to prioritise significant features. This SR module improves the resolution of the volume, restoring it to a level close to the original.

The reconstructed volume, denoted as $y$, is compared to the original $x$ using an $L_1$ loss function to assess and optimise reconstruction quality. The entire system is trained using the Adam optimiser with a learning rate of 0.0015 for 5,000 epochs on an NVIDIA A4000 16GB GPU with CUDA support in the PyTorch framework. All source codes and results are available at https://github.com/asheibanifard/EndtoEndCompression.

### 4.2 Trade-off architecture

**4.2.1 3D downsampling module.**    The Downsampling module does not require training. This implies that the downsampling scale is per set without consideration of the final result quality. We select three downsampling scales of 1/2, 1/4, and 1/8 in our experiments. It is necessary to test the performance of the proposed architecture at three downsampling scales, particularly the INR module. Table 2 presents a comprehensive comparison of reconstruction results for different downsampling scales, illustrating the effectiveness of our proposed architecture in maintaining a high reconstruction quality across various compression levels. It can be noted that decreasing the downsampling scales does not significantly degenerate the quality

**Table 2. Performance of the INR module and the whole end-to-end architecture.** (The upper row shows the performance of a single SIREN and the lower row shows that of the whole end-to-end architecture).

| INR module | | | | | |
|---|---|---|---|---|---|
| Scale | Avg Bitrate ↓ | Avg CR (%)↑ | Avg PSNR ↑ | Avg SSIM ↑ | Avg #Para ↓ |
| 1/2 | 5.21 | 83.71 | 36.96 | 0.95 | 42711 |
| 1/4 | 5.21 | 83.71 | 51.48 | 1.00 | 42711 |
| 1/8 | 5.21 | 83.71 | 67.34 | 1.00 | 42711 |
| Whole end-to-end Architecture | | | | | |
| Scale | Avg Bitrate ↓ | Avg CR (%)↑ | Avg PSNR ↑ | Avg SSIM ↑ | Avg #Para ↓ |
| 1/2 | 5.743 | 82.052 | 38.001 | 0.956 | 47048.0 |
| 1/4 | 6.655 | 79.200 | 38.381 | 0.953 | 54524.0 |
| 1/8 | 10.062 | 68.553 | 39.462 | 0.961 | 82436.0 |

of the reconstruction. Additionally, non-standard sampling scales like 1/3, 1/5, or 1/7 would introduce unnecessary complexity and inconsistencies without offering meaningful improvements, making them less suitable for the architecture's goals. Thus, these three downsampling scales are acceptable.

**4.2.2 3D INR module.** We opt for the SIREN model [9] as our INR module, focusing on two primary aspects of the SIREN structure: the number of layers and the number of neurons per layer. The goal is to use a compact SIREN model to enhance the compression rate (CR). We experiment with various configurations of the SIREN model, altering the layer count and neuron count per layer, as detailed in Table 3. We find that a SIREN network with 3 layers and between 30 and 230 neurons per layer offers satisfactory performance, especially for small volume data inputs, while substantially cutting down on GPU memory usage. Furthermore, we compare the performance of a single SIREN model against our proposed architecture, as shown in Table 2. The notable benefit is a dramatic reduction in GPU memory consumption while maintaining comparable reconstruction quality. Additionally, using more than 230 neurons per layer increases the model's capacity to represent detailed structures but leads to diminishing returns in terms of reconstruction quality. Beyond 230 neurons, the gains in PSNR and SSIM are marginal, while the computational cost and GPU memory usage increase significantly. This increased complexity does not translate into substantial improvements in performance, making the additional computational overhead unjustified. Thus, we prefer the SIREN model with 3 layers in the INR module.

**4.2.3 3D super-resolution module.** We utilise the SRNO [11] for the SR module due to its compact size, as evidenced by the average number of parameters of deep networks in Table 2. We also compare our end-to-end architecture with cutting-edge methods [32–37]. Table 8 reveals that (1) the SR module performs effectively, as our architecture, using a 3-layer SIREN, matches the reconstruction quality of a standalone 5-layer SIREN; and (2) our architecture surpasses other state-of-the-art image compression methods in terms of PSNR and SSIM.

**4.2.4 Find a trade-off architecture by trade-off point approach.** To find the trade-off point for our proposed architecture, firstly, our proposed architecture is tested in terms of all combinations of NC, DS and SN, which is presented separately in Table 4 with 4 channels of feature extraction in the SRNO model, Table 5 with 8 channels of feature extraction in the SRNO model, and Table 6 with 16 channels of feature extraction in the SRNO model. The trade-off point of the proposed architecture is then calculated using Eq 8, that is, the trade-off point (NC = 4, DS = 1/2, SN = 30). At the trade-off point, the PSNR upper bound is around 38,

**Table 3. Average values for different INR layers and neurons.**

| Layers | Neurons | Bitrate(bpp) ↓ | CR (%)↑ | PSNR ↑ | SSIM ↑ | #Para ↓ |
|---|---|---|---|---|---|---|
| 3 | 30 | 0.245 | 99.233 | 31.081 | 0.767 | 2011 |
| 3 | 50 | 0.653 | 97.959 | 32.205 | 0.804 | 5351 |
| 3 | 70 | 1.256 | 96.074 | 34.550 | 0.903 | 10291 |
| 3 | 90 | 2.055 | 93.579 | 35.637 | 0.923 | 16831 |
| 3 | 110 | 3.048 | 90.474 | 36.610 | 0.942 | 24971 |
| 3 | 130 | 4.237 | 86.759 | 37.862 | 0.960 | 34711 |
| 3 | 150 | 5.621 | 82.433 | 38.389 | 0.964 | 46051 |
| 3 | 170 | 7.201 | 77.497 | 38.626 | 0.965 | 58991 |
| 3 | 190 | 8.976 | 71.950 | 39.954 | 0.975 | 73531 |
| 3 | 210 | 10.946 | 65.793 | 39.553 | 0.974 | 89671 |
| 3 | 230 | 13.112 | 59.026 | 40.934 | 0.981 | 107411 |
| 4 | 30 | 0.359 | 98.878 | 29.008 | 0.627 | 2941 |
| 4 | 50 | 0.964 | 96.986 | 32.472 | 0.825 | 7901 |
| 4 | 70 | 1.863 | 94.178 | 34.814 | 0.902 | 15261 |
| 4 | 90 | 3.054 | 90.455 | 36.450 | 0.937 | 25021 |
| 4 | 110 | 4.539 | 85.817 | 37.251 | 0.951 | 37181 |
| 4 | 130 | 6.316 | 80.262 | 39.872 | 0.974 | 51741 |
| 4 | 150 | 8.386 | 73.793 | 41.887 | 0.984 | 68701 |
| 4 | 170 | 10.750 | 66.407 | 42.395 | 0.986 | 88061 |
| 4 | 190 | 13.406 | 58.107 | 42.738 | 0.988 | 109821 |
| 4 | 210 | 16.355 | 48.890 | 43.586 | 0.989 | 133981 |
| 4 | 230 | 19.597 | 38.758 | 44.335 | 0.991 | 160541 |
| 5 | 30 | 0.473 | 98.523 | 30.846 | 0.781 | 3871 |
| 5 | 50 | 1.276 | 96.013 | 32.469 | 0.799 | 10451 |
| 5 | 70 | 2.470 | 92.282 | 38.391 | 0.964 | 20231 |
| 5 | 90 | 4.054 | 87.331 | 38.456 | 0.963 | 33211 |
| 5 | 110 | 6.029 | 81.159 | 40.037 | 0.976 | 49391 |
| 5 | 130 | 8.395 | 73.766 | 41.990 | 0.985 | 68771 |
| 5 | 150 | 11.151 | 65.152 | 42.953 | 0.988 | 91351 |
| 5 | 170 | 14.298 | 55.318 | 42.284 | 0.986 | 117131 |
| 5 | 190 | 17.836 | 44.263 | 43.366 | 0.989 | 146111 |
| 5 | 210 | 21.764 | 31.987 | 44.676 | 0.991 | 178291 |
| 5 | 230 | 26.083 | 18.491 | 44.616 | 0.991 | 213671 |
| 6 | 30 | 0.586 | 98.169 | 30.077 | 0.774 | 4801 |
| 6 | 50 | 1.587 | 95.041 | 36.246 | 0.935 | 13001 |
| 6 | 70 | 3.076 | 90.387 | 39.529 | 0.974 | 25201 |
| 6 | 90 | 5.054 | 84.207 | 40.204 | 0.977 | 41401 |
| 6 | 110 | 7.520 | 76.501 | 41.779 | 0.984 | 61601 |
| 6 | 130 | 10.474 | 67.270 | 41.733 | 0.984 | 85801 |
| 6 | 150 | 13.916 | 56.512 | 43.195 | 0.988 | 114001 |
| 6 | 170 | 17.847 | 44.229 | 42.773 | 0.988 | 146201 |
| 6 | 190 | 22.266 | 30.420 | 44.301 | 0.991 | 182401 |
| 6 | 210 | 27.173 | 15.084 | 42.274 | 0.984 | 222601 |
| 6 | 230 | 32.568 | -1.777 | 43.441 | 0.988 | 266801 |

**Table 4. The results of our proposed architecture with 4 channels of shallow feature extractor in SR module.**

| Scale | # Neurons | Bitrate(bpp) ↓ | CR(%)↑ | PSNR(db) ↑ | SSIM ↑ | #Para ↓ | GPU memory(GB) ↓ |
|---|---|---|---|---|---|---|---|
| 1/2 | 30 | 0.775 | 97.578 | 33.885 | 0.885 | 6348 | 1.366 |
| 1/2 | 50 | 1.183 | 96.304 | 35.211 | 0.915 | 9688 | 1.395 |
| 1/2 | 70 | 1.786 | 94.420 | 36.853 | 0.947 | 14628 | 1.426 |
| 1/2 | 90 | 2.584 | 91.925 | 37.682 | 0.961 | 21168 | 1.456 |
| 1/2 | 110 | 3.578 | 88.820 | 38.547 | 0.969 | 29308 | 1.484 |
| 1/2 | 130 | 4.767 | 85.104 | 38.968 | 0.972 | 39048 | 1.512 |
| 1/2 | 150 | 6.151 | 80.779 | 38.867 | 0.974 | 50388 | 1.540 |
| 1/2 | 170 | 7.730 | 75.842 | 39.419 | 0.973 | 63328 | 1.570 |
| 1/2 | 190 | 9.505 | 70.296 | 39.603 | 0.976 | 77868 | 1.599 |
| 1/2 | 210 | 11.476 | 64.139 | 39.319 | 0.970 | 94008 | 1.629 |
| 1/2 | 230 | 13.641 | 57.372 | 39.665 | 0.976 | 111748 | 1.661 |
| 1/4 | 30 | 1.520 | 95.250 | 33.221 | 0.858 | 12452 | 1.319 |
| 1/4 | 50 | 1.928 | 93.976 | 33.985 | 0.892 | 15792 | 1.322 |
| 1/4 | 70 | 2.531 | 92.091 | 34.503 | 0.916 | 20732 | 1.331 |
| 1/4 | 90 | 3.329 | 89.597 | 34.789 | 0.921 | 27272 | 1.335 |
| 1/4 | 110 | 4.323 | 86.491 | 34.753 | 0.915 | 35412 | 1.333 |
| 1/4 | 130 | 5.512 | 82.776 | 34.825 | 0.921 | 45152 | 1.349 |
| 1/4 | 150 | 6.896 | 78.450 | 35.080 | 0.923 | 56492 | 1.348 |
| 1/4 | 170 | 8.476 | 73.514 | 35.001 | 0.924 | 69432 | 1.347 |
| 1/4 | 190 | 10.250 | 67.967 | 34.977 | 0.919 | 83972 | 1.347 |
| 1/4 | 210 | 12.221 | 61.810 | 35.300 | 0.927 | 100112 | 1.354 |
| 1/4 | 230 | 14.386 | 55.043 | 35.393 | 0.922 | 117852 | 1.355 |
| 1/8 | 30 | 7.481 | 76.622 | 40.991 | 0.977 | 61284 | 1.313 |
| 1/8 | 50 | 7.889 | 75.348 | 36.212 | 0.909 | 64624 | 1.314 |
| 1/8 | 70 | 8.492 | 73.463 | 40.873 | 0.977 | 69564 | 1.312 |
| 1/8 | 90 | 9.290 | 70.969 | 38.934 | 0.965 | 76104 | 1.315 |
| 1/8 | 110 | 10.284 | 67.863 | 40.995 | 0.979 | 84244 | 1.319 |
| 1/8 | 130 | 11.473 | 64.148 | 40.799 | 0.978 | 93984 | 1.316 |
| 1/8 | 150 | 12.857 | 59.822 | 40.150 | 0.975 | 105324 | 1.318 |
| 1/8 | 170 | 14.437 | 54.886 | 39.587 | 0.974 | 118264 | 1.316 |
| 1/8 | 190 | 16.211 | 49.339 | 39.866 | 0.973 | 132804 | 1.318 |
| 1/8 | 210 | 18.182 | 43.182 | 38.954 | 0.966 | 148944 | 1.317 |
| 1/8 | 230 | 20.347 | 36.415 | 39.094 | 0.960 | 166684 | 1.319 |

the SSIM upper bound is around 0.94, and the CR upper bound is around 76.6%, as shown in Table 7. This is a good setting for the proposed architecture, as it reaches a high compression rate and good quality for reconstruction.

Moreover, it is further illustrated by Eq 9. We show the three sets of marginal distributions concerning dimensions (NC, DS, SN), in Figs 3–5, respectively. If CR is decreased, the SIREN size (SN) or channel number (NC) can be increased. However, the reconstruction quality (i.e. PSNR or SSIM) shows a slight improvement. Thus, enlarging the model size or channel number will not significantly improve reconstruction quality. Additionally, compared to other existing approaches in Table 8, our architecture excels in maintaining a low Bitrate(bpp), ensuring that the compressed file size is significantly smaller. Our results (PSNR and SSIM) are still comparable with those of the "3D-VOI-OMLSVD [34]". Fig 6 further shows the reconstructed slices of volume data.

**Table 5. The results of our proposed network with 8 channels of shallow feature extractor in SR module.**

| Scale | # Neurons | Bitrate(bpp) ↓ | CR(%)↑ | PSNR(db) ↑ | SSIM ↑ | #Para ↓ | GPU memory(GB) ↓ |
|---|---|---|---|---|---|---|---|
| 1/2 | 30 | 1.688 | 94.727 | 32.829 | 0.825 | 13824 | 1.330 |
| 1/2 | 50 | 2.095 | 93.452 | 36.009 | 0.922 | 17164 | 1.360 |
| 1/2 | 70 | 2.698 | 91.568 | 37.112 | 0.945 | 22104 | 1.387 |
| 1/2 | 90 | 3.497 | 89.073 | 38.502 | 0.964 | 28644 | 1.422 |
| 1/2 | 110 | 4.490 | 85.968 | 39.373 | 0.975 | 36784 | 1.451 |
| 1/2 | 130 | 5.679 | 82.253 | 39.127 | 0.974 | 46524 | 1.479 |
| 1/2 | 150 | 7.063 | 77.927 | 39.365 | 0.971 | 57864 | 1.506 |
| 1/2 | 170 | 8.643 | 72.990 | 41.106 | 0.982 | 70804 | 1.537 |
| 1/2 | 190 | 10.418 | 67.444 | 40.133 | 0.979 | 85344 | 1.562 |
| 1/2 | 210 | 12.388 | 61.287 | 38.495 | 0.976 | 101484 | 1.595 |
| 1/2 | 230 | 14.554 | 54.520 | 40.149 | 0.976 | 119224 | 1.625 |
| 1/4 | 30 | 3.171 | 90.091 | 35.348 | 0.886 | 25976 | 1.277 |
| 1/4 | 50 | 3.579 | 88.817 | 36.051 | 0.917 | 29316 | 1.281 |
| 1/4 | 70 | 4.182 | 86.932 | 37.443 | 0.941 | 34256 | 1.289 |
| 1/4 | 90 | 4.980 | 84.438 | 36.178 | 0.935 | 40796 | 1.291 |
| 1/4 | 110 | 5.974 | 81.332 | 38.016 | 0.948 | 48936 | 1.295 |
| 1/4 | 130 | 7.163 | 77.617 | 37.564 | 0.946 | 58676 | 1.307 |
| 1/4 | 150 | 8.547 | 73.291 | 38.021 | 0.948 | 70016 | 1.307 |
| 1/4 | 170 | 10.126 | 68.355 | 36.991 | 0.944 | 82956 | 1.307 |
| 1/4 | 190 | 11.901 | 62.808 | 38.424 | 0.950 | 97496 | 1.310 |
| 1/4 | 210 | 13.872 | 56.651 | 36.755 | 0.942 | 113636 | 1.311 |
| 1/4 | 230 | 16.037 | 49.884 | 36.696 | 0.942 | 131376 | 1.319 |
| 1/8 | 30 | 15.038 | 53.006 | 45.056 | 0.990 | 123192 | 1.273 |
| 1/8 | 50 | 15.446 | 51.732 | 44.628 | 0.989 | 126532 | 1.272 |
| 1/8 | 70 | 16.049 | 49.847 | 45.004 | 0.990 | 131472 | 1.276 |
| 1/8 | 90 | 16.847 | 47.353 | 44.226 | 0.987 | 138012 | 1.272 |
| 1/8 | 110 | 17.841 | 44.247 | 45.528 | 0.992 | 146152 | 1.277 |
| 1/8 | 130 | 19.030 | 40.532 | 43.596 | 0.985 | 155892 | 1.276 |
| 1/8 | 150 | 20.414 | 36.206 | 44.410 | 0.990 | 167232 | 1.276 |
| 1/8 | 170 | 21.994 | 31.270 | 43.539 | 0.986 | 180172 | 1.276 |
| 1/8 | 190 | 23.769 | 25.723 | 43.650 | 0.988 | 194712 | 1.277 |
| 1/8 | 210 | 25.739 | 19.566 | 42.061 | 0.981 | 210852 | 1.277 |
| 1/8 | 230 | 27.904 | 12.799 | 41.435 | 0.978 | 228592 | 1.275 |

Additionally, Fig 7 shows a steady optimisation process over 5000 epochs, with continuous improvements in reconstruction accuracy and structural similarity. The PSNR curve exceeds 40 dB, indicating high reconstruction quality with minimal error. The SSIM curve approaches 0.96, demonstrating the model's effectiveness in preserving perceptual and structural fidelity. The steady decrease in the loss function, alongside the PSNR and SSIM improvements, confirms effective convergence. These results, consistent with the final performance metrics in Table 8, highlight the architecture's ability to balance compression efficiency and high-quality reconstruction, making it ideal for medical imaging.

**Remark:** The proposed trade-off point approach serves as a pragmatic optimisation strategy. In the context of the compression problem, it is essential to balance various requirements, including downsampling scales, INR module size, SR module structure, etc., rather than over-emphasising one or two factors. The trade-off point approach addresses this challenge by elegantly optimising the parameters involved.

**Table 6. The results of our proposed network with 16 channels of shallow feature extractor in SR module.**

| Scale | # Neurons | Bitrate(bpp) ↓ | CR(%)↑ | PSNR(db) ↑ | SSIM ↑ | #Para ↓ | GPU memory(GB) ↓ |
|---|---|---|---|---|---|---|---|
| 1/2 | 30 | 5.095 | 84.079 | 34.469 | 0.858 | 41736 | 1.366 |
| 1/2 | 50 | 5.502 | 82.805 | 37.843 | 0.947 | 45076 | 1.396 |
| 1/2 | 70 | 6.105 | 80.920 | 39.609 | 0.965 | 50016 | 1.424 |
| 1/2 | 90 | 6.904 | 78.426 | 39.634 | 0.967 | 56556 | 1.457 |
| 1/2 | 110 | 7.897 | 75.320 | 40.585 | 0.979 | 64696 | 1.486 |
| 1/2 | 130 | 9.086 | 71.605 | 38.699 | 0.975 | 74436 | 1.514 |
| 1/2 | 150 | 10.471 | 67.279 | 41.600 | 0.982 | 85776 | 1.546 |
| 1/2 | 170 | 12.050 | 62.343 | 39.853 | 0.969 | 98716 | 1.568 |
| 1/2 | 190 | 13.825 | 56.796 | 41.416 | 0.981 | 113256 | 1.601 |
| 1/2 | 210 | 15.795 | 50.639 | 41.327 | 0.977 | 129396 | 1.632 |
| 1/2 | 230 | 17.961 | 43.872 | 39.053 | 0.977 | 147136 | 1.660 |
| 1/4 | 30 | 8.055 | 74.829 | 35.132 | 0.891 | 65984 | 1.272 |
| 1/4 | 50 | 8.462 | 73.555 | 38.964 | 0.941 | 69324 | 1.279 |
| 1/4 | 70 | 9.065 | 71.671 | 40.981 | 0.965 | 74264 | 1.290 |
| 1/4 | 90 | 9.864 | 69.176 | 40.182 | 0.965 | 80804 | 1.289 |
| 1/4 | 110 | 10.857 | 66.071 | 40.705 | 0.969 | 88944 | 1.292 |
| 1/4 | 130 | 12.046 | 62.355 | 42.462 | 0.979 | 98684 | 1.298 |
| 1/4 | 150 | 13.431 | 58.029 | 42.580 | 0.981 | 110024 | 1.307 |
| 1/4 | 170 | 15.010 | 53.093 | 41.079 | 0.970 | 122964 | 1.308 |
| 1/4 | 190 | 16.785 | 47.546 | 42.322 | 0.979 | 137504 | 1.306 |
| 1/4 | 210 | 18.755 | 41.389 | 41.177 | 0.974 | 153644 | 1.311 |
| 1/4 | 230 | 20.921 | 34.622 | 41.987 | 0.979 | 171384 | 1.313 |
| 1/8 | 30 | 31.734 | 0.830 | 49.798 | 0.998 | 259968 | 1.270 |
| 1/8 | 50 | 32.142 | -0.444 | 48.750 | 0.996 | 263308 | 1.271 |
| 1/8 | 70 | 32.745 | -2.328 | 48.238 | 0.995 | 268248 | 1.271 |
| 1/8 | 90 | 33.543 | -4.823 | 48.497 | 0.997 | 274788 | 1.271 |
| 1/8 | 110 | 34.537 | -7.928 | 46.755 | 0.993 | 282928 | 1.280 |
| 1/8 | 130 | 35.726 | -11.644 | 48.436 | 0.997 | 292668 | 1.275 |
| 1/8 | 150 | 37.110 | -15.970 | 50.539 | 0.999 | 304008 | 1.277 |
| 1/8 | 170 | 38.690 | -20.906 | 48.897 | 0.996 | 316948 | 1.273 |
| 1/8 | 190 | 40.465 | -26.453 | 48.730 | 0.997 | 331488 | 1.275 |
| 1/8 | 210 | 42.435 | -32.610 | 46.971 | 0.994 | 347628 | 1.277 |
| 1/8 | 230 | 44.601 | -39.377 | 47.476 | 0.996 | 365368 | 1.277 |

## 5 Conclusion and future work

In this paper, we proposed an innovative architecture that integrates available deep-learning techniques with a focus on compressing volume data while maintaining high reconstruction fidelity. One notable aspect of our approach is the utilisation of emerging deep learning technologies, which have witnessed rapid development in recent years. We emphasised the

**Table 7. Our proposed architecture's trade-off point.**

| Marginal values | NC = 4 | DS = 1/2 | SN = 30 |
|---|---|---|---|
| 1/PSNR | 0.02598 | 0.02681 | 0.02694 |
| $1 - |SSIM|$ | 0.04287 | 0.05491 | 0.09239 |
| $1 - CR$ | 0.23398 | 0.25709 | 0.25887 |

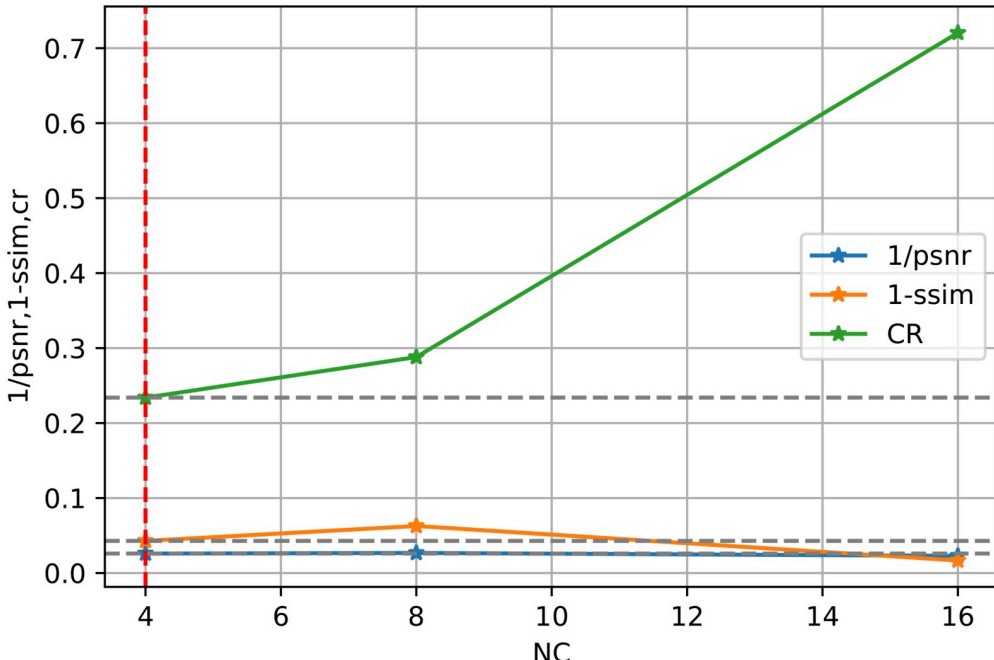

**Fig 3. Illustrates the trade-off point for the number of channels (NC) in the SR module concerning the performance metrics, 1/PSNR, 1-SSIM, and 1-CR.** The red dashed lines indicate the intersection where the optimal trade-off is achieved, balancing compression efficiency and reconstruction quality.

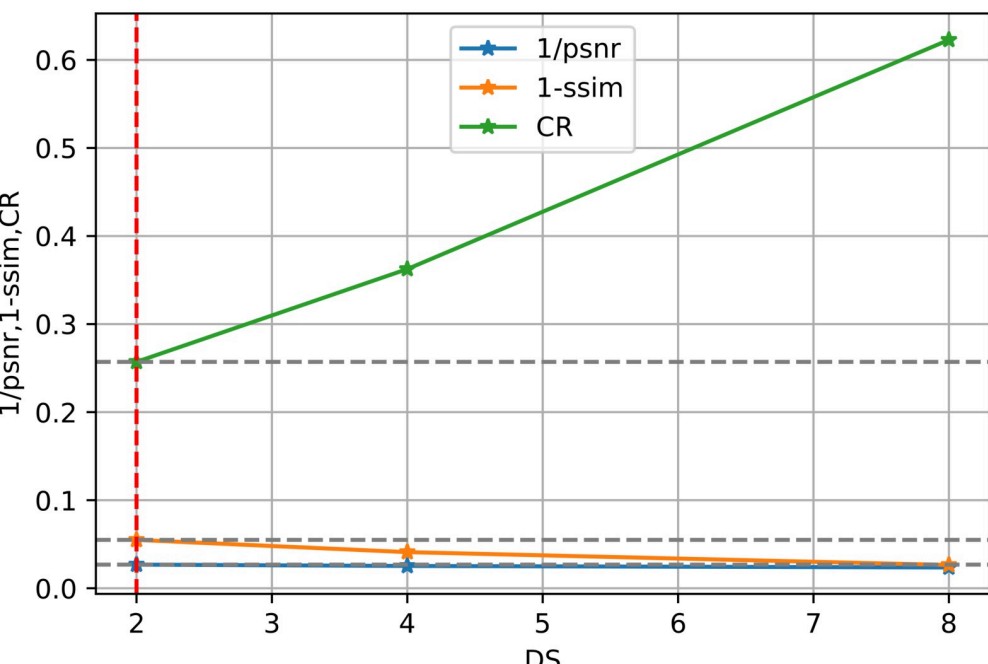

**Fig 4. The trade-off point for the downsampling scale (DS) is based on the performance metrics, 1/PSNR, 1-SSIM, and 1-CR.** The red dashed lines highlight where the downsampling scale achieves an optimal balance between compression rate and reconstruction accuracy.

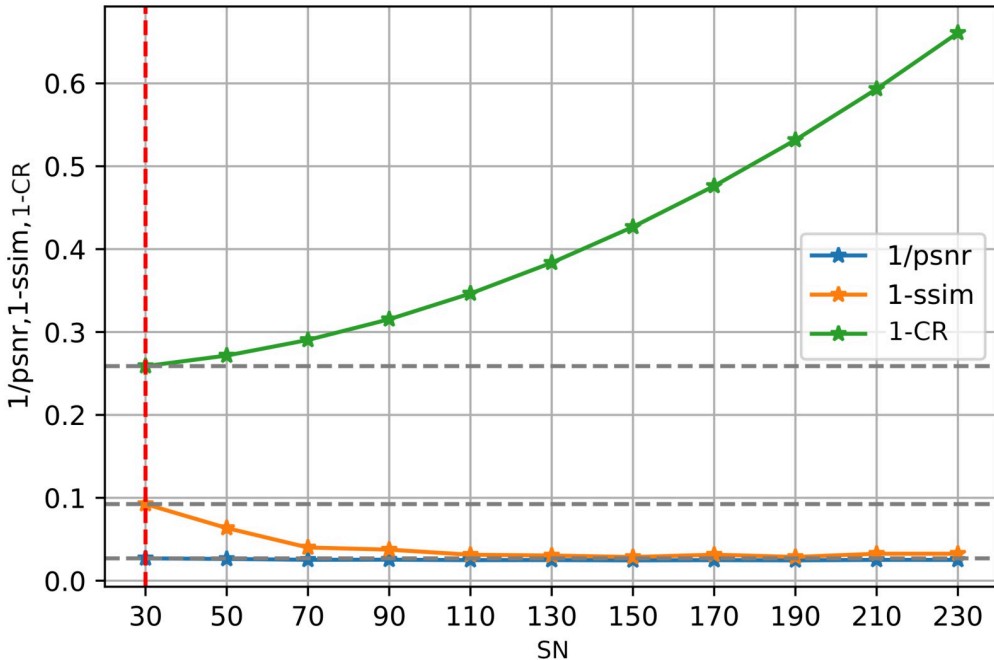

**Fig 5. The trade-off point for the number of neurons (SN) in the SIREN model, plotted against the performance metrics, 1/PSNR, 1-SSIM, and 1-CR.** The red dashed lines indicate the optimal configuration of neurons in the SIREN model for achieving high reconstruction quality with minimal compression loss.

importance of carefully considering various factors such as network architecture, computational efficiency, and reconstruction accuracy when designing and implementing the end-to-end solution. To this end, we proposed the end-to-end network architecture for volume data compression and developed the trade-off approach to determine optimal settings for individual modules, which is a practical method to balance performance considerations in the context of medical visualisation tasks.

## 5.1 Limitations

**5.1.1 Generalisation to diverse medical datasets.** Applying the proposed end-to-end architecture to various volume datasets requires significant retraining time for each dataset individually, as there is no fine-tuning strategy in place to speed up this process.

**Table 8. Comparison of our techniques with other state-of-the-art methods in terms of PSNR and SSIM in volume reconstruction.**

| Method | avg PSNR ↑ | avg SSIM ↑ | CR(%)↑ | Bitrate(bpp) ↓ | GPU(GB) ↓ |
|---|---|---|---|---|---|
| Single SIREN [9] | 40.008 | 0.947 | 67.062 | 10.348 | 3.390 |
| Devadoss et al. [33] | 34.1098 | - | 78.16 | 4.580 | - |
| MVAR [32] | 40.050 | - | 90.00 | - | - |
| 3D-VOI-OMLSVD [34] | **42.04** | **0.978** | 89.17 | 2.54 | - |
| aiWave-heavy [36] | 39.00 | - | - | 2.5 | - |
| Block CS [35] | 30.86 | 0.7489 | 50.00 | - | - |
| EZW with Haar [37] | 30.15 | - | 40.31 | - | - |
| Our Architecture | 40.052 | 0.961 | **97.578** | **0.775** | **0.769** |

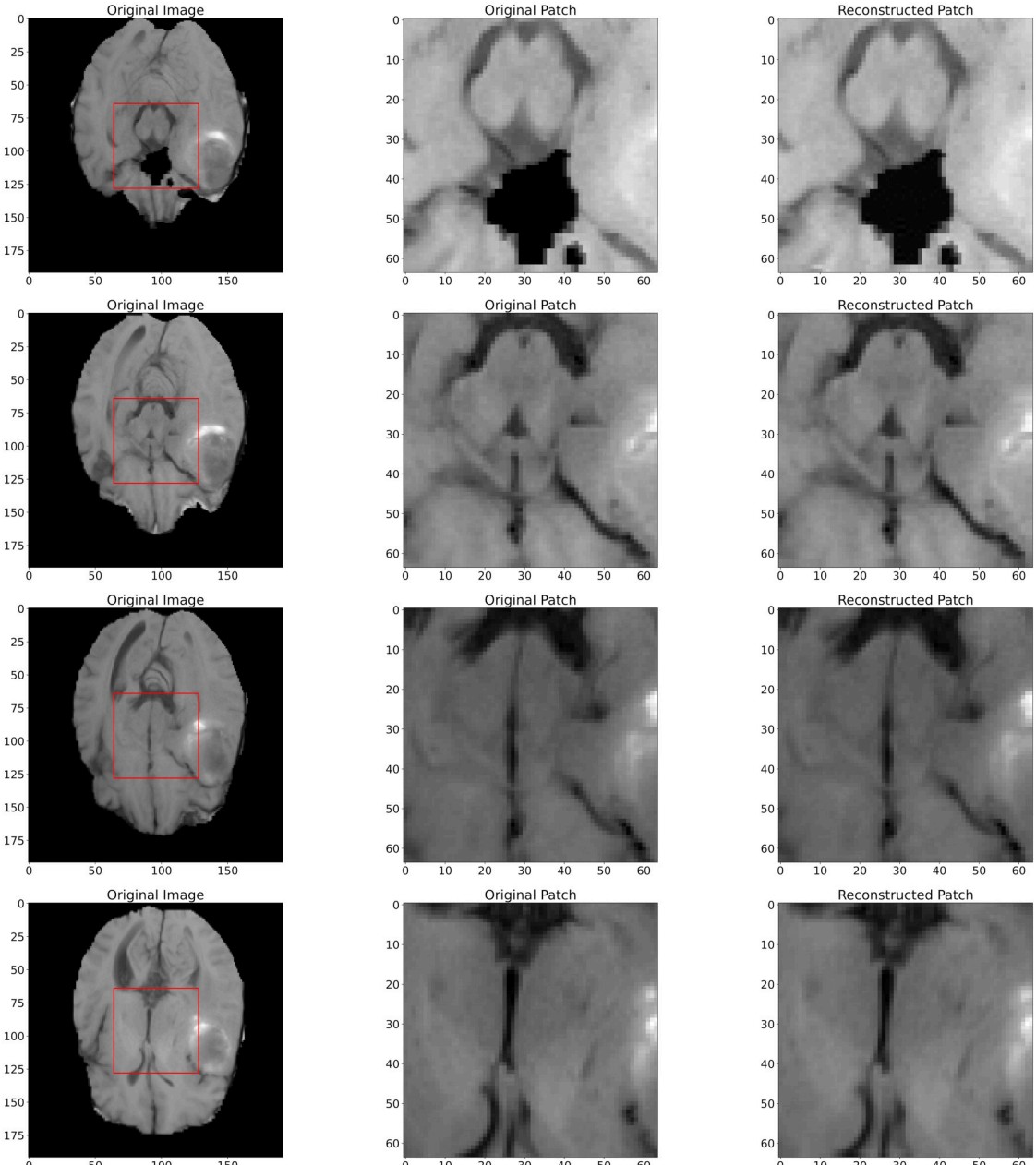

**Fig 6. The Left column shows the different original slices of the volume with sizes of (155, 240, 240); the middle column shows the labelled patches of the slices with sizes of (64, 64, 64); the right column shows the reconstructed patches by our architecture.**

**5.1.2 Time complexity of trade-off point approach.** The trade-off point method necessitates sampling the model's performance across different architecture settings, which is highly time-consuming.

## 5.2 Future work

Beyond the realm of compression, visualising over-large medical volume data through real-time rendering is meaningful. Compression with rendering could enable real-time

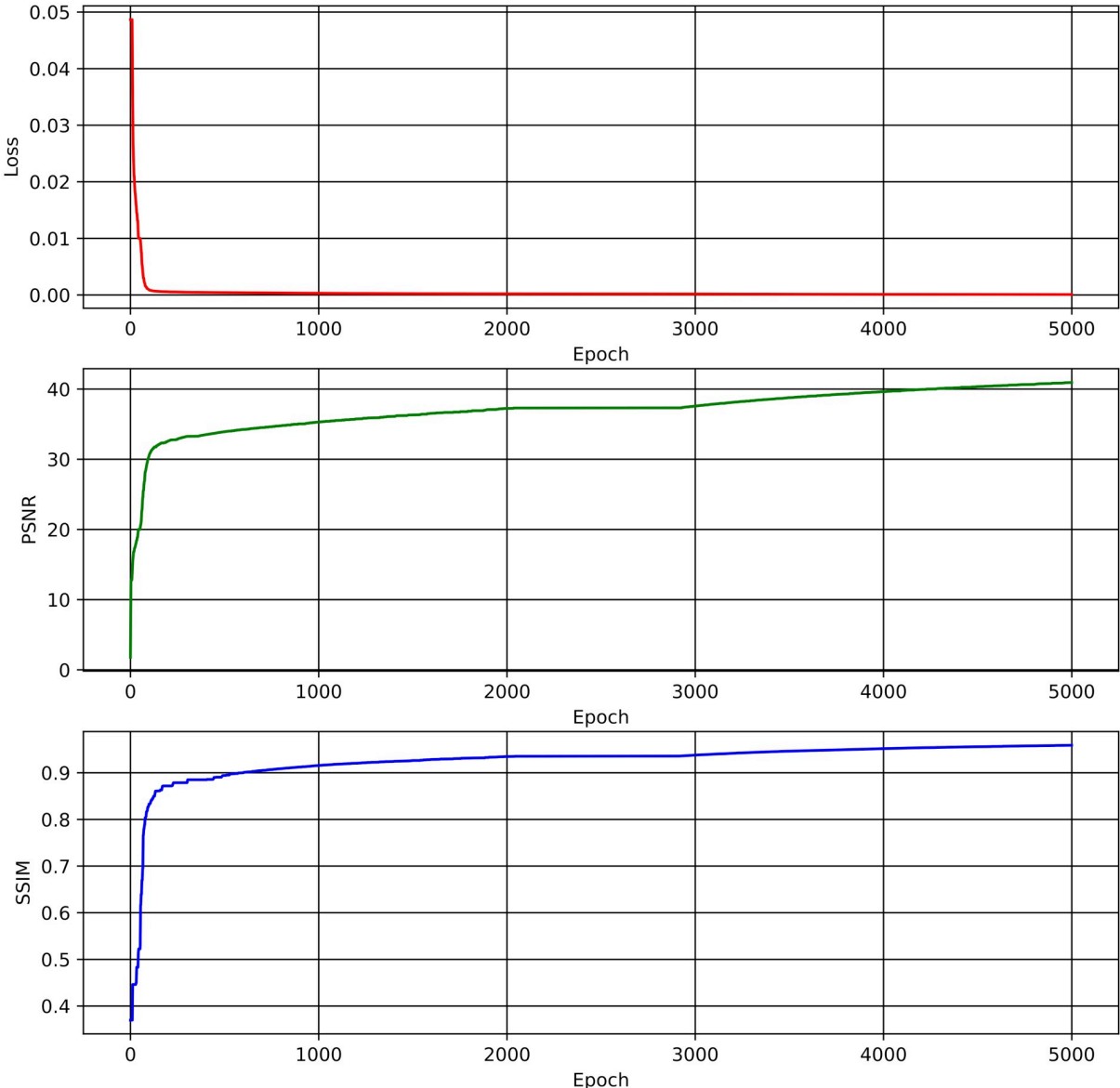

**Fig 7. Training procedure of the architecture according to the trade-off point setting.**

visualisation of such over-large volume data. In future work, we intend to focus on volume-rendering techniques that leverage implicit neural representations. This research direction shows significant promise for advancements in the field of visualisation.

## Author Contributions

**Conceptualization:** Hongchuan Yu.

**Funding acquisition:** Hongchuan Yu, Jian J. Zhang.

**Investigation:** Armin Sheibanifard, Zongcai Ruan.

**Methodology:** Hongchuan Yu, Zongcai Ruan.

**Project administration:** Hongchuan Yu, Jian J. Zhang.

**Resources:** Jian J. Zhang.

**Software:** Armin Sheibanifard.

**Supervision:** Hongchuan Yu.

**Writing – original draft:** Armin Sheibanifard, Zongcai Ruan.

**Writing – review & editing:** Hongchuan Yu.

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
