## [Decision Letter · Decision Letter 0]

21 May 2024

PONE-D-24-12715An End-to-End Implicit Neural Representation Network for Medical Volume DataPLOS ONE

Dear Dr. Yu,

Thank you for submitting your manuscript to PLOS ONE. After careful consideration, we feel that it has merit but does not fully meet PLOS ONE’s publication criteria as it currently stands. Therefore, we invite you to submit a revised version of the manuscript that addresses the points raised during the review process.

We look forward to receiving your revised manuscript.

Kind regards,

Rossana Mastrandrea

Academic Editor

PLOS ONE

Journal Requirements:

   "EU Horizon Marie Skłodowska-Curie Action;

Grant No. 101130271;

Title: Affective Computing Models: from Facial Expression to Mind-Reading"

Reviewers' comments:

Reviewer's Responses to Questions

**Comments to the Author**

1. Is the manuscript technically sound, and do the data support the conclusions?

Reviewer #1: Yes

Reviewer #2: Partly

2. Has the statistical analysis been performed appropriately and rigorously? 

Reviewer #1: I Don't Know

Reviewer #2: Yes

3. Have the authors made all data underlying the findings in their manuscript fully available?

Reviewer #1: Yes

Reviewer #2: No

4. Is the manuscript presented in an intelligible fashion and written in standard English?

Reviewer #1: Yes

Reviewer #2: Yes

5. Review Comments to the Author

Reviewer #1: A preliminary theoretical analysis of the proposed methodology motivates its rationale. Also, this manuscript has many problems, such as structure, contribution, proposed method, etc. So, In my opinion, this is not suitable for publication.

Reviewer #2: The issues are listed in the following:

1. The professional English editing is recommended. The authors should get editing help from someone with full professional proficiency in English.

2. What is the main difference or importance of the proposed methods and the other state-of-the-arts?

3. The abstract should reflect the contributions of the manuscript. I suggest rewriting it.

4. More recent references in the context of this study need to be added and discussed in manuscript. For example, “A Secure Visual Framework for Multi-index Privacy Protection Evaluation in Networks, doi: 10.1016/j.dcan. 2022.05.007.”

5. The Conclusion section should point out the potential disadvantages and possible future research directions of the manuscript. How this work can be extended in future?

6. PLOS authors have the option to publish the peer review history of their article (what does this mean?). If published, this will include your full peer review and any attached files.

Reviewer #1: No

Reviewer #2: No

---

## [Author Response · Author response to Decision Letter 0]

11 Jul 2024

We have submitted response letter.

---

## [Decision Letter · Decision Letter 1]

17 Sep 2024

PONE-D-24-12715R1An End-to-End Implicit Neural Representation Architecture for Medical Volume DataPLOS ONE

Dear Dr. Yu,

Thank you for submitting your manuscript to PLOS ONE. After careful consideration, we feel that it has merit but does not fully meet PLOS ONE’s publication criteria as it currently stands. Therefore, we invite you to submit a revised version of the manuscript that addresses the points raised during the review process.

We look forward to receiving your revised manuscript.

Kind regards,

Xiyu Liu

Academic Editor

PLOS ONE

Reviewers' comments:

Reviewer's Responses to Questions

**Comments to the Author**

1. If the authors have adequately addressed your comments raised in a previous round of review and you feel that this manuscript is now acceptable for publication, you may indicate that here to bypass the “Comments to the Author” section, enter your conflict of interest statement in the “Confidential to Editor” section, and submit your "Accept" recommendation.

Reviewer #1: (No Response)

Reviewer #2: All comments have been addressed

2. Is the manuscript technically sound, and do the data support the conclusions?

Reviewer #1: (No Response)

Reviewer #2: Yes

3. Has the statistical analysis been performed appropriately and rigorously? 

Reviewer #1: (No Response)

Reviewer #2: Yes

4. Have the authors made all data underlying the findings in their manuscript fully available?

Reviewer #1: (No Response)

Reviewer #2: Yes

5. Is the manuscript presented in an intelligible fashion and written in standard English?

Reviewer #1: (No Response)

Reviewer #2: Yes

6. Review Comments to the Author

Reviewer #1: This manuscript proposes an innovative architecture that integrates available deep-learning techniques focusing on compressing volume data while maintaining high reconstruction fidelity. Adequate revisions to the following points should be undertaken to justify the recommendation for publication.

The authors should clearly state the limitations of the proposed method in other real applications.

The abstract section is fragile. Please rewrite it, explain the result obtained and contribution, improve a proposed method, and delete unnecessary information.

Proofread the manuscript carefully to eliminate any grammatical errors or typos and ensure clarity and coherence in writing. Additionally, adhere to the formatting and style guidelines specified by the target journal or publication venue to enhance the professionalism of the manuscript.

I suggest the authors add a table at the end of the literature review and compare the reviewed papers to clarify the research gap better.

Please write your contribution to this paper in the Introduction section.

Expand the critical results in the conclusion. Focus on the main developments in the finale. Also, write the main contributions in the conclusion.

Numerical results are good enough, but more explanations are required to analyze each figure presented.

The simulation section needs to be more detailed. The authors should provide more information about the data they employed and the simulation process.

Please Change the “conclusion” section to “ Conclusion and Future Work” and write future work.

All figures are of low quality, so please improve all of them.

Good luck

Reviewer #2: The issues are listed in the following:

1. The professional English editing is recommended. The authors should get editing help from someone with full professional proficiency in English.

2. The abstract contains a large amount of information. It is recommended to simplify the language and highlight the innovative points and main results of the study. It is suggested to update some higher quality or newer literature. For example, the introduction of convolutional neural networks can be referenced as "A Novel Centralized Federated Deep Fuzzy Neural Network with Multi-objectives. Neural Architecture Search for Epistatic Detection, DOI: 10.1109 / TFUZZ. 2024.3369944 ', Fourier Transform technique introduction can be quoted "An IoMT enabled deep learning framework for automatic detection of fetal QRS: A solution to remote prenatal care, DOI10.1016 / j.j ksuci. 2022.07.002.

3. The figures included in the manuscript are not sufficiently clear. High-resolution images should be provided to ensure that all details are visible and can be thoroughly examined by readers.

4. The introduction should provide a clearer overview of the motivation, objectives, and main contributions of the study. Additionally, the limitations of existing technologies and how your work addresses these gaps should be briefly discussed.

5. Please outline your main contributions more clearly in the introduction and conclusion sections. This should include the specific innovations of your proposed end-to-end architecture and the advantages over traditional methods.

6. Your proposed end-to-end architecture requires a more detailed description.

7. The results section requires more details to substantiate your claims. For instance, for the performance evaluation of each module, more quantitative data and visualizations should be provided.

8. The selection of the dataset, experimental setup, and evaluation metrics need more detailed descriptions to ensure the reproducibility and validity of the results.

9. Please ensure that all relevant and up-to-date literature is cited and that the citation format adheres to the journal's requirements.

10. Discuss in more detail within the text how your work relates to existing technologies. This includes a comparison of your work with the current state-of-the-art and where your work offers improvements.

7. PLOS authors have the option to publish the peer review history of their article (what does this mean?). If published, this will include your full peer review and any attached files.

Reviewer #1: No

Reviewer #2: No

---

## [Author Response · Author response to Decision Letter 1]

25 Oct 2024

- Response to reviewer #1: 

1. The proposed method’s limitations have been stated in the conclusion in the revision [349-356]

2. The abstract has been rewritten in the revision

3. The grammatical errors or typos have been corrected in the revision and Guidelines were checked

4. The suggested table has been added as Table 1 in the revision

5. The contributions mentioned before, but they have been expanded for clarification in the revision.[31-37]

6. The conclusion has been rewritten accordingly in the revision [338-348]

7. In Find a trade-off architecture by trade-off point approach subsection, we added Fig 7 to show the training procedure of the end to end architecture. Moreover, we also rewrote the Materials and Experimental results Section to explain each figure and Table in details in the revision. [313-330]

8. The required explanations about simulation process have been added in the Materials and Experimental Results section. Data and implementation setup subsection has been rewritten in the revision. [234-260]

9. It has been changed in the revision [337]

10. The PACE tool standardised the images according to the guidelines. But this time for assurance, the original figures' configurations and quality were adapted to the guidelines with a maximum resolution of 600 dpi. 

- Response to reviewer #2: 

1. The revision has been proofread to correct errors.

2. These missed reference have been cited in the revision. [11-12]

3. The PACE tool standardised the images according to the guidelines. For assurance, the original high quality figures were uploaded instead in the revision.

4. The introduction has been rewritten accordingly and for clarification, table 1 has been added to show the gaps in the revision. 

5. The contributions have been explicitly mentioned in introduction and conclusion in the revision.[31-37][338-348]

6. The Proposed End-to-End architecture subsection has been updated accordingly and Data and implementation setup subsection has been updated as well in the revision. [100-104] 

7. In Find a trade-off architecture by trade-off point approach subsection, we added Fig 7 to show the training procedure of the end to end architecture. Moreover, we also rewrote the Materials and Experimental results Section to explain each figure and Table in details in the revision.[265-268][272-274][287-293]

8. More details have been added to the Data and implementation setup subsection in the revision. [231-234]

9. The citations and their format have been reviewed in the revision.

10. Table 7 shows the comparison with SOTA and has been updated in both caption and description in text in the revision.[319-321]

---

## [Decision Letter · Decision Letter 2]

19 Nov 2024

An End-to-End Implicit Neural Representation Architecture for Medical Volume Data

PONE-D-24-12715R2

Dear Dr. Yu,

We’re pleased to inform you that your manuscript has been judged scientifically suitable for publication and will be formally accepted for publication once it meets all outstanding technical requirements.

Kind regards,

Xiyu Liu

Academic Editor

PLOS ONE

Additional Editor Comments (optional):

Reviewers' comments:

Reviewer's Responses to Questions

**Comments to the Author**

1. If the authors have adequately addressed your comments raised in a previous round of review and you feel that this manuscript is now acceptable for publication, you may indicate that here to bypass the “Comments to the Author” section, enter your conflict of interest statement in the “Confidential to Editor” section, and submit your "Accept" recommendation.

Reviewer #1: All comments have been addressed

Reviewer #2: All comments have been addressed

2. Is the manuscript technically sound, and do the data support the conclusions?

Reviewer #1: Yes

Reviewer #2: Yes

3. Has the statistical analysis been performed appropriately and rigorously? 

Reviewer #1: Yes

Reviewer #2: Yes

4. Have the authors made all data underlying the findings in their manuscript fully available?

Reviewer #1: Yes

Reviewer #2: Yes

5. Is the manuscript presented in an intelligible fashion and written in standard English?

Reviewer #1: Yes

Reviewer #2: Yes

6. Review Comments to the Author

Reviewer #1: The authors have addressed all the issues according to my previous comments. The related work has been enriched, and the indistinct description and deficient analysis have been further refined. More discussions have also been added. This paper has been revised thoroughly to reach the standard for publication. Consequently, I advise you to accept this paper.

Reviewer #2: All of my questions have been addressed and have been revised as required for the proposed publication in this journal.

7. PLOS authors have the option to publish the peer review history of their article (what does this mean?). If published, this will include your full peer review and any attached files.

Reviewer #1: No

Reviewer #2: No

---

## [Editor Report · Acceptance letter]

20 Dec 2024

PONE-D-24-12715R2 

PLOS ONE

Dear Dr. Yu, 

I'm pleased to inform you that your manuscript has been deemed suitable for publication in PLOS ONE. Congratulations! Your manuscript is now being handed over to our production team.

Kind regards, 

on behalf of

Professor Xiyu Liu 

Academic Editor

PLOS ONE